# *Skeletonema marinoi* Extracts and Associated Carotenoid Fucoxanthin Downregulate Pro-Angiogenic Mediators on Prostate Cancer and Endothelial Cells

**DOI:** 10.3390/cells12071053

**Published:** 2023-03-30

**Authors:** Luana Calabrone, Valentina Carlini, Douglas M. Noonan, Marco Festa, Cinzia Ferrario, Danilo Morelli, Debora Macis, Angelo Fontana, Luigi Pistelli, Christophe Brunet, Clementina Sansone, Adriana Albini

**Affiliations:** 1IRCCS MultiMedica, 20138 Milan, Italy; 2Department of Biotechnology and Life Sciences, University of Insubria, 21100 Varese, Italy; 3IRCCS Istituto Europeo di Oncologia IEO, 20141 Milan, Italy; 4Institute of Biomolecular Chemistry, Italian National Research Council (CNR), 80078 Pozzuoli, Italy; 5Department of Biology, Università di Napoli “Federico II”, 80126 Napoli, Italy; 6Stazione Zoologica Anton Dohrn, Villa Comunale, 80121 Napoli, Italy

**Keywords:** diatoms, fucoxanthin, prostate, endothelial, prevention

## Abstract

The exploration of natural preventive molecules for nutraceutical and pharmaceutical use has recently increased. In this scenario, marine microorganisms represent an underestimated source of bioactive products endowed with beneficial effects on health that include anti-oxidant, anti-inflammatory, differentiating, anti-tumor, and anti-angiogenic activities. Here, we tested the potential chemopreventive and anti-angiogenic activities of an extract from the marine coastal diatom *Skeletonema marinoi* Sarno and Zingone (*Sm*) on prostate cancer (PCa) and endothelial cells. We also tested one of the main carotenoids of the diatom, the xanthophyll pigment fucoxanthin (Fuco). Fuco from the literature is a potential candidate compound involved in chemopreventive activities. *Sm* extract and Fuco were able to inhibit PCa cell growth and hinder vascular network formation of endothelial cells. The reduced number of cells was partially due to growth inhibition and apoptosis. We studied the molecular targets by qPCR and membrane antibody arrays. Angiogenesis and inflammation molecules were modulated. In particular, Fuco downregulated the expression of Angiopoietin 2, CXCL5, TGFβ, IL6, STAT3, MMP1, TIMP1 and TIMP2 in both prostate and endothelial cells. Our study confirmed microalgae-derived drugs as potentially relevant sources of novel nutraceuticals, providing candidates for potential dietary or dietary supplement intervention in cancer prevention approaches.

## 1. Introduction

The marine environment comprises a wide variety of systems, thus providing extremely high biodiversity and boundless chemical richness. Natural compounds have always attracted a great deal of attention due to their potential effects in promoting health. So far, however, only about twenty drugs based on marine-origin compounds have been approved for clinical purposes [1,2].

Currently, most marine-derived products are used mostly as nutraceuticals with interesting beneficial effects on human health due to their content of active ingredients deploying pathology prevention-associated functions, including anti-inflammatory, anti-oxidant, anti-microbial, anti-angiogenic, and pro-apoptotic activities [3,4].

Chemoprevention is the use of agents (vitamins, drugs, or nutritional supplements) to reduce, inhibit, delay, or reverse carcinogenesis [5]. Angiogenesis is an essential physiological and vital process in growth and normal development with the formation of new blood vessels from pre-existing vessels, but if abnormal, it can contribute to many pathologies, including tumor growth and metastasis. Angioprevention consists of inhibiting tumor angiogenesis in a preventive setting [5]. In this context, angioprevention can be considered as a chemopreventive approach denoted to prevent cancer-associated angiogenesis [5,6].

Numerous in vitro and in vivo studies have established the potential chemopreventive properties of various biologically active phytochemicals isolated from natural products [7,8]. In particular, compounds derived from the marine environment [9,10,11,12,13], including microalgal species and their products [14,15,16,17,18], can be of great interest.

Recently, microalgae have received attention for being rich in nutrients and for their ability to produce bioactive compounds (polysaccharides, proteins, bioactive peptides, polyunsaturated lipids, carotenoids and pigments, polyphenolic compounds, minerals) and have been shown to be sources of therapeutic agents valuable in the treatment of different human diseases and that can be employed in cancer chemoprevention [19,20,21,22,23].

Among the biodiversity of microalgae, diatoms are one of the most widespread and diversified group that populate all aquatic environments [24]. Diatoms represent promising candidates for biotechnological applications, such as bioremediation, as well as for nutraceutical and pharmaceutical purposes [18,25,26,27,28,29] due to their primary metabolites (carotenoids, vitamins, polyphenols, and polyunsaturated fatty acids [PUFA]). *Skeletonema marinoi* (*Sm*), a temperate coastal marine diatom, has been recently proposed as a valid source of bioactive molecules. Diatoms contain several moieties of interest for human health, among them carotenoids, including fucoxanthin (Fuco) [11,28,30,31,32,33], diatoxanthin, and other pigments [34,35,36].

Fuco represents the most abundant carotenoid contained in diatoms such as *Sm* and can be also isolated from brown seaweeds for human consumption [33]. Numerous studies have reported that Fuco has biological properties, acting as an anti-obesity, anti-diabetes, anti-oxidant, and anti-inflammatory molecule, and it also has cardiovascular and cerebrovascular protective effects [37]. Therefore, it is used as both a pharmaceutical and nutritional ingredient to prevent and treat chronic diseases.

Prostate cancer (PCa) represents the most commonly diagnosed malignancy in men and among the 3rd leading cause of male cancer deaths worldwide in 2020 [38]. Substantial evidence suggests that chronic inflammation and angiogenesis contribute to tumor initiation, metastasis, and progression [39,40,41,42,43,44]. From first diagnosis to progression of PCa, there is a large time span during which natural drugs could be administered in a secondary prevention setting, likely avoiding recurrences and metastatic disease [45,46,47,48].

Diverse naturally occurring compounds and dietary supplements (such as polyphenols, flavonoids, carotenoids, etc.) have been reported to be endowed with chemopreventive and angiopreventive activities in PCa [45,46] by targeting multiple pathways, thus interfering with cancer insurgence, progression, and metastasis [5,38,40,49,50,51]. These compounds exhibit anti-proliferative, anti-inflammatory, anti-angiogenic, anti-oxidant, and pro-apoptotic activities [5,38,40,49,50,51]. The major features of these agents are represented by low toxicity to normal cells of the host and high tolerability over long-term administration [5,38,40,49,50,51].

Despite recent advances in diagnosis and treatment, PCa remains one of the leading causes of cancer-related deaths in men worldwide [38]. Currently, available treatments can increase 5-year survival in the early stages of PCa, but the metastatic disease is still difficult to manage [52]. Epidemiological and clinical studies strongly support the association between nutrition and development or progression of major cancers, including PCa [53]. Strategies aimed to prevent PCa onset are urgently needed and PCa is considered an ideal cancer type for chemopreventive approaches that also include supplement interventions. Additionally, angiogenesis appears relevant in PCa. Restricted data are available comparing the effects of microalgal extracts and their single pigments on endothelial cells [18,54,55], particularly in an oncological setting.

Although prostate cancer (PCa) is a valid candidate for chemopreventive and nutritional interventions, only about ten/fifteen papers are available from the last 10 years studying the effects of Fuco on PCa. However, Fuco’s potential anti-cancer properties against other cancers have been reported [33].

Here, we focused on the properties of an extract from the diatom *Sm* and the algal pigment Fuco, which is highly present in *Sm*, to inhibit PCa cells in vitro and analyzed the cellular mechanisms involved in a potential chemopreventive approach. Evaluation of the effects of the microalgal extract and pure pigment on endothelial cell function could be related to their ability to prevent angiogenic switch, hence reducing tumor progression. Several studies have investigated the effects of microalgae, including their extracts and pigments, on cancer [14,25,26,56,57,58,59,60,61,62,63,64,65,66,67,68], but much less has been reported about marine pigments in cancer-related angiogenesis [69,70,71,72].

The aim of this study was to evaluate in vitro the possible chemopreventive effect of *Sm* extract and Fuco towards PCa and, furthermore, their anti-angiogenic properties on endothelial cells. We found that the *Sm* extract and pure Fuco were able to inhibit PCa cell growth and inhibited vascular network formation and vascular mimicry of endothelial cells. Fuco downregulated the expression of genes involved in angiogenesis and inflammation processes, such as Angiopoietin 2, CXCL5, TGFβ, IL6, STAT3, MMP1, TIMP1 and TIMP2, in both prostate and endothelial cells. Our results demonstrate promising chemopreventive and angiopreventive activities of *Sm* extracts, containing high doses of Fuco, on PCa cell lines. Both *Sm* extract and Fuco could effectively inhibit proliferation, promote apoptosis, and also suppress vascular mimicry formation in PCa cells. Moreover, they appear able to reduce the migration ability and expression of genes involved in inflammatory and angiogenic processes in TNFα-stimulated HUVECs.

## 2. Materials and Methods

### 2.1. Microalgal Biomass and Pure Pigments

The marine diatom *Sm* was grown as previously described by Pistelli et al. (2021) [73]. *Sm* extract was then prepared as follows: dried biomass was suspended in 500 μL of methanol and shaken vigorously for 2 minutes (min). The solution was sonicated for 1 min with a micro tip at 20% output on ice (S-250A Branson Ultrasonic) and then the tubes were centrifuged at 3900× *g* for 15 min at 4 °C. The supernatants were then dried by rotary evaporation at 37 °C and the dried methanolic extracts were resuspended in dimethyl sulfoxide (DMSO, 0.1%) and human cell culture medium. The quantities of carotenoids, particularly Fuco and diatoxanthin, were measured (Appendix A). For experimental purposes, pure fucoxanthin was purchased from Sigma-Aldrich (St. Louis, MO, USA) and was dissolved in ethanol (EtOH) or DMSO.

### 2.2. Cell Line Culture and Maintenance

The in vitro assays were performed using the DU145, LNCaP, and 22Rv1 PCa cell lines purchased from the American Type Culture Collection (ATCC) and human umbilical vein endothelial cells (HUVEC, Lonza). DU145 and 22Rv1 cells were grown in RPMI 1640 medium supplemented with 10% fetal bovine serum (FBS, Gibco, Thermo Fisher Scientific, Waltham, MA, USA), 2 mM L-glutamine, and 1% penicillin/streptomycin (Pen/Strep, Sigma-Aldrich, St. Louis, MO, USA) and maintained at 37 °C under 5% CO_2_. LNCaP cells were maintained in RPMI 1640 medium, as described above, and supplemented with 1% sodium pyruvate. HUVECs were cultured in endothelial cell basal medium (EBM™, Lonza) supplemented with endothelial cell growth medium (EGM™SingleQuots™, Lonza), 10% FBS, 2 mM L-glutamine, and 1% Pen/Strep. Cell lines were maintained at 37 °C under 5% CO_2_. HUVECs were used between passages 2–7. Cells were regularly tested for mycoplasma infection.

### 2.3. Generation of Conditioned Media

Conditioned media (CM), for subsequent array analysis, were obtained from the HUVEC and DU145 cell lines. Briefly, 0.3 × 10^6^ cells were seeded into 6-well plates (Corning) in complete medium (EBM™ for HUVEC and RPMI for DU145) at 37 °C under 5% CO_2_. When cells reached 80% of confluence, HUVECs were stimulated for 1 h with TNFα (10 ng/mL), then both cell lines were washed for 30 min with medium without serum and were starved for 24 hours (h) for HUVECs and 48 h for DU145 cells in 1 mL/well serum-free medium, supplemented or not with Fuco (20 µg/mL). Finally, CM were collected, pulling two wells per condition, and residual cells and debris were discarded by centrifugation.

DU145 and LNCaP cells were cultured as above in T75 or T175 flasks to 80–85% confluence, rinsed with 1× phosphate-buffered saline (PBS), then cultured in their respective medium without FBS. CM were harvested after 48 h and filtered through the Amicon^®^ Ultra-15 device (50 kDa, Merck Millipore, Darmstadt, Germany) to remove cells and debris. All CMs collected were stored at −80 °C.

### 2.4. Detection of Cell Proliferation

#### 2.4.1. Crystal Violet Viability Assay

Cell viability was determined using crystal violet staining solution. Briefly, 5 × 10^3^ cells from the DU145 and LNCaP cell lines were seeded in 96-well plates and, after adhesion, treated with *Sm* (10 and 100 µg/mL) and Fuco (from 0.5 to 50 µg/mL). DMSO, at the corresponding higher concentration, was used as the control for *Sm* whereas EtOH, at the corresponding higher concentration, was used as the control for Fuco. Complete RPMI medium alone or with 0.2% saponin (Sigma-Aldrich) was used as the positive and negative controls, respectively. At the pre-fixed timepoints, after a rapid wash in PBS and fixation in 4% paraformaldehyde (PFA) for 10 min at room temperature (RT), 50 µL of crystal violet staining solution (Sigma Aldrich) was added to each well and left for 20 min at RT. Then, plates were gently washed in tap water and air-dried overnight at RT. Cell-retained crystal violet was dissolved in 100 µL of crystal violet elution buffer (50% EtOH and 0.1% acetic acid in distilled water) and the absorbance was immediately recorded at a wavelength of 595 nanometers (nm) using the microplate spectrophotometer SpectraMax M2. Experiments were performed in triplicate and repeated three times. After this first assay, the same experiment was performed on DU145 and LNCaP cells adding 100 µg/mL to the range of Fuco doses (0.5, 1, 5, 10, and 50 µg/mL). EtOH, at the corresponding concentration, was used as the control. Then, the protocol was performed as described above. Selected experiments were also conducted on a third cell line, 22Rv1 PCa cells.

#### 2.4.2. MTT Cell Growth Assay

Cell proliferation was assessed by MTT (3-(4,5-dimethylthiazol-2-yl)-2,5-diphenyltetrazolium bromide; Sigma-Aldrich) colorimetric assay. DU145 PCa cells (5 × 10^3^ cells/well) were seeded in 96-well plates and, after adhesion, treated with *Sm* (1, 10, and 100 µg/mL) for 24, 48, and 72 h. The solvent DMSO at the corresponding dilutions was used as the control. Complete RPMI medium alone or with 0.2% saponin was used as the positive and negative controls, respectively. Effects of treatment were tested at 24, 48, and 72 h at increasing concentrations of *Sm*.

At the pre-fixed timepoints, treatments were removed, and cells were incubated for 3 h at 37 °C under 5% CO_2_ in 100 µL of fresh complete RPMI medium supplemented with 10 µL of MTT (stock solution: 5 mg/mL). MTT was then removed, and formazan crystals were dissolved using 100% DMSO for 15 min at 350 rpm. The absorbance was immediately recorded at a wavelength of 570 nm using the microplate spectrophotometer SpectraMax M2 (Molecular Devices, Sunnyvale, CA, USA). Experiments were performed in three replicates and repeated twice.

HUVECs (5 × 10^3^ cells/well) were seeded into 96-well plates. Following cell adhesion, corresponding complete media with decimal dilutions of *Sm* (1, 10, and 100 µg/mL) were added for 24, 48, and 72 h. DMSO dissolved in the corresponding complete medium was used as the vehicle control. Medium alone or with 0.2% saponin was used as the negative and positive controls, respectively. HUVEC growth was determined by crystal violet assay. Briefly, 5 × 10^3^ HUVECs were seeded into 96-well plates and, following cell adhesion, treated with *Sm* (0.5, 1, 5, 10, 20, 50, and 100 µg/mL) and Fuco (0.5, 1, 5, 10, and 50 µg/mL). DMSO and EtOH dissolved in complete RPMI were used as the vehicle control for *Sm* and the pigment, respectively. Complete medium alone or with 0.2% saponin was used as the negative and positive controls, respectively. Then, the protocol was performed as described above for PCa cells.

### 2.5. Determination of Apoptosis by Flow Cytometer

DU145 (1.5 × 10^6^ cells/well) and 22vR1 (1.5 × 10^6^ cells/well) cells were treated with *Sm* (100 and 200 µg/mL) or Fuco (20 and 100 µg/mL) for 24 and 48 h. DMSO, at the corresponding dilutions, was used as the vehicle control, 40% EtOH was used as the positive pro-apoptotic control. Induction of apoptosis was determined by propidium iodide (PI, 1 μg/mL) (Sigma Aldrich) and Annexin-V-APC (Immunotools GmbH, Friesoythe, Germany) staining, followed by flow cytometry analysis using a BD FACS CantoII flow cytometer. Flow data were analyzed using FACSDiva 6.1.2 software (Becton Dickinson (BD) Biosciences, San Jose, CA, USA) and FlowLogic (Miltenyi Biotec, Bergisch Gladbach, Germany) software. Experiments were performed in duplicate.

### 2.6. Prostate Cancer Cell Vascular Mimicry

The vasculogenic mimicry assay was performed on the DU145 PCa cell line. A 96-well plate (from two to five wells per treatment) was pre-coated with Matrigel (BD Biosciences) Growth Factor Reduced Matrix (10 mg/mL, 70 µL/well; Corning, AZ, USA) that was allowed to polymerize for 30 min at 37 °C under 5% CO_2_. Then, 5 × 10^4^ DU145 cells were added to each well and treated with *Sm* (100 and 200 µg/mL) and Fuco (20 µg/mL and 100 µg/mL). DMSO, at the corresponding dilutions, was used as the vehicle control for both *Sm* and Fuco. RPMI medium with 10% FBS or FBS-free was used as the positive and negative controls, respectively. All treatments were performed in 10% FBS RPMI medium to induce vasculogenic mimicry. Cells were incubated at 37 °C under 5% CO_2_ for 24 h. One photo was captured per well under an inverted light microscope (Zeiss Axio Observe A1) associated with a digital camera (Zeiss Axiocam MRm) at 5× magnification following 24 h of treatments. Tubules in each field were imaged and the number of tubules from 3–5 random fields in each well were averaged. Captured images were analyzed using ImageJ software and the Angiogenesis Analyzer tool [74] to measure relevant parameters for efficient vasculogenic mimicry, such as total segment length and total master segment length, and the mean values were used for the analysis. For statistical analyses, four to six wells were seeded per condition. Experiments were repeated twice.

### 2.7. Endothelial Tube Formation Assay

The effect of *Sm* and pigment on endothelial cell ability to form a capillary-like structure on the basement membrane matrix was assessed in vitro using the morphogenesis assay. HUVECs (5 × 10^5^ cells/well) were grown in complete EBM-2 medium and when they were 80% confluent, they were serum-starved overnight. A 96-well plate, pre-chilled at −20 °C, was carefully filled with 50 μL/well of liquid Matrigel at 4 °C and then polymerized for 1 h at 37 °C.

To evaluate the effects on HUVEC tube formation, cells were suspended in 100 μL of EBM-2 complete medium alone or with treatment and layered on top of the polymerized Matrigel. Positive and negative controls received 10% FBS or serum-free EBM-2 medium, respectively. Following 6 or 24 h of incubation, one picture per well (center of the well and four cardinal points) was captured at 5× magnification under an inverted light microscope (Zeiss microscope) equipped with a digital camera (Zeiss Axiocam MRm) and quantified using ImageJ software. The Angiogenesis Analyzer tool was used to measure relevant parameters for tubulogenic efficiency, including number (Nb) of master segments, Nb meshes, Nb nodes, and total mesh area. For statistical analyses, four to six wells were seeded per condition. Experiments were repeated twice.

### 2.8. Detection of Cell Migration

#### 2.8.1. Boyden Chamber and Transwell Migration Assays

The effect of Fuco on the migration ability of HUVECs was investigated using a modified Boyden chamber system. HUVECs were incubated with starvation medium overnight and then seeded (4 × 10^4^ cells/well) in the upper compartment of the Boyden chamber in EBM-2 serum-free medium containing Fuco (20 μg/mL) or EtOH (vehicle control). A 10 μm pore-size polycarbonate filter coated with fibronectin (2 μg/mL, Sigma Aldrich) was used as the interface between the upper and lower chambers. EBM-2 medium addition with 10% FBS or DU145 CM was placed in the lower chamber to induce cell migration along the chemoattractant gradient. Serum-free and complete media were used as the negative and positive controls for migration, respectively. Following 6 h of incubation (migration) at 37 °C under 5% CO_2_, non-migrated cells were removed from the interior of the insert using a cotton swab, the Boyden chamber was disassembled, and the filter, containing migrated cells, was recovered, washed in PBS, and fixed in 4% PFA for 10 min at RT.

We also evaluated HUVEC migration with the transwell assay using 8 μm pore-size cell culture inserts (Millipore, Burlington, MA, USA). HUVECs were starved overnight, seeded (6 × 10^4^ cells/well) in the upper chamber of the transwell culture insert, and treated with Fuco (20 μg/mL) or EtOH (vehicle control) in EBM-2 serum-free medium. In the lower chamber, complete EBM-2 medium with 10% FBS was added to direct the cell migration. Serum-free and complete EBM-2 media were used as the negative (FBS-) and positive (FBS+) controls, respectively. Following 6 h of incubation, non-migrated cells were removed as described above, while cells migrated on the underside of the transwell culture insert were washed and fixed in 4% PFA.

In both assays, migrated cells were stained with crystal violet solution and photographed using an inverted light microscope (Zeiss Axio Observer A1 microscope) associated with a digital camera (Zeiss Axiocam MRm) at 10× magnification. The average number of migrated cells was obtained by counting the cells in different fields in a double-blind manner, with four filters per condition, using ImageJ software (Wayne Rasband, National Institutes of Health, Bethesda, MD, USA).

#### 2.8.2. Scratch Wound Healing Migration Assay

The scratch wound assay was used to assess the ability of Fuco (20 µg/mL) to inhibit migration of HUVECs and DU145 cells in vitro. Cells were seeded at 5 × 10^4^/well in 12-well plates and incubated at 37 °C under 5% CO_2_ until and they reached full confluence. Cell monolayers were then scratched with a 20–200 µL micropipette tip to create a wound of ~1 mm width. Cells were then washed twice with PBS to remove debris and detached cells and incubated with 2 mL of 1% FBS fresh medium containing 20 µg/mL of Fuco. For each scratch, images were acquired at 0 and 24 h for HUVECs and at 0, 24, and 48 h for DU145 cells using an inverted light microscope (Leica DM IL) equipped with a digital camera (Leica MC190 HD) at 4× magnification. Wound closure was checked by observing the narrowing of the area between scratch borders at the selected time points.

### 2.9. Effects of CM from PCa Cells on Endothelial Cell Morphogenesis or Matrigel Morphogenesis Assay

The ability of Fuco to limit angiogenesis induction by DU145 cells via soluble factor collected in CM was evaluated thanks to the ability of HUVECs to form capillary-like structures on a 3D basement membrane matrix.

A 96-well plate, pre-chilled at −20 °C, was carefully coated with 10 mg/mL of reduced growth factor Matrigel (BD Biosciences) at 4 °C using a pre-chilled pipette tip. The Matrigel was then polymerized for 1 h at 37 °C. HUVECs at 15 × 10^3^ cells/well were layered on top of the polymerized Matrigel, received the collected CM (50 µg total protein) from DU145 cells in FBS-free EBM-2 medium, and were incubated for 6 and 24 h. The positive control received 10% FBS EBM-2 medium. Capillary-like network formation was observed under a Zeiss microscope associated with a Nikon camera (Axio Observer A1, Zeiss), at 10× magnification and quantified using ImageJ software with the Angiogenesis Analyzer tool. Each experiment was performed in triplicate on HUVECs either at basal level or activated with TNFα (10 ng/mL) and repeated twice.

### 2.10. Quantitative Real-Time PCR (qPCR)

A total of 1 × 10^6^ cells of DU145 PCa were seeded into p100 plates and, following adhesion (80–90% confluence), were treated with *Sm* (100 µg/mL) and Fuco (20 µg/mL). DMSO for *Sm* and EtOH for Fuco, at the corresponding dilutions, were used as the vehicle control. After 6 and 24 h, total RNA was extracted following standard procedures, as described below.

HUVECs were seeded in 6-well plates (2.5 × 10^5^ cells/well) in complete endothelial cell basal medium. We investigated the potential effects of *Sm* and marine pigments on the gene expression of HUVECs in a model of inflammation induced by TNFα. Following cell adhesion, HUVECs were incubated in the presence or absence of TNFα (10 ng/mL) for 1 h and then fresh complete medium with *Sm* (100 µg/mL) and Fuco (20 µg/mL) was added for 6 h at 37 °C. EtOH, at the corresponding dilution, was used as the control. Not treated cells (NT) received complete medium. Then, total RNA was extracted following standard procedures, as described below.

Briefly, total RNA was extracted using the TRIzol method, following separation with chloroform precipitation of RNA with isopropanol (Sigma-Aldrich). The RNA pellet was washed twice with 75% EtOH and resuspended in nuclease-free water. The RNA concentration was determined using a Nanodrop Spectrophotometer ND-1000 (Thermo Fisher Scientific). Reverse transcription was performed using the SuperScript VILO cDNA synthesis kit (Thermo Fisher Scientific), starting from 1000 ng of total RNA.

Quantitative real-time qPCR was performed using the SYBR Green Master Mix (Applied Biosystems, Waltham, MA, USA) with the QuantStudio 6 Flex Real-Time PCR System (Applied Biosystems). All reactions were performed in triplicate and repeated three times. The relative gene expression was expressed relative to TNFα-stimulated cells or solvent control and normalized to the housekeeping gene. Cycles up to 30 were taken into account.

Gene expression analysis was performed using the primers shown in Appendix A. Primers were designed using the NCBI Primer BLAST tool and purchased from Integrated DNA Technologies (IDT, Coralville, IA, USA). Data were analyzed using MeV Software.

### 2.11. Secretome Analysis of HUVEC and DU145 Prostate Cells (Human Angiogenesis Antibody Array)

TNFα was added to HUVECs at 10 ng/mL and incubated for 1 h, before proceeding with Fuco (20 µg/mL) treatment and EtOH, at the corresponding dilution, was used as the vehicle control. To analyze the secretome, CM were recovered from cells cultured under FBS-free medium during the last 24 h of culture for HUVECs and 48 h for DU145 cells. For antibody array experiments, pooled CM from two independent biological replicates were used. The RayBio^®^ Human Angiogenesis Antibody Array kit (RayBiotech) was used following the manufacturer’s instructions. This method is a dot-based assay enabling the detection and comparison of 20 angiogenesis-specific cytokines.

The images of blotted proteins were acquired by Alliance Q9 ATOM (Uvitec Cambridge). Alliance software (UVITEC, Cambridge, UK) was used to quantify the arrays. An average signal was calculated of the two spots representing each protein, followed by background subtraction of the clear area of the membrane. The data were analyzed using the RayBio Antibody Array Analysis Tool (Ray Biotech), according to the manufacturer’s protocol. Experiments were repeated twice.

### 2.12. Statistical Analysis

Data from experiments including at least three technical replicates per treatment were analyzed using GraphPad Prism™ software (GraphPad Software, Inc., San Diego, CA, USA) and expressed as the mean ± standard error of mean (SEM). One-way analysis of variance (one-way ANOVA) followed by Tukey post hoc test was used to analyze differences between treatments. Two groups of unpaired data were analyzed using the Student’s *t*-test. Data were considered statistically significant for *p*-values < 0.05.

## 3. Results

### 3.1. Effect of S. marinoi Extract and Fucoxanthin on PCa Cell Proliferation by Crystal Violet and MTT Assays

The effect of *Sm* treatment on DU145, LNCaP, and 22Rv1 cells proliferation was assessed by crystal violet experiments. Saponin was used as the control for apoptosis. As shown in Figure 1A, *Sm* (100 µg/mL) was active in reducing DU145 cell number and significantly decreased the number of LNCaP cells at 10 μg/mL after 72 h of treatment. The data were confirmed by the MTT results at increasing doses (1, 10, and 100 μg/mL). *Sm* at 100 μg/mL significantly decreased DU145 cell viability (*p* ≤ 0.0001) while LNCaP cell viability was significantly decreased with treatment of *Sm* at 10 and 100 μg/mL (*p* ≤ 0.001 and *p* ≤ 0.0001, respectively) after 48 h of treatment. Doses of 1 and 10 µg/mL of *Sm* were not sufficiently active in reducing DU145 cell proliferation (Figure 1B). The data for DU145 cells after 24 and 48 h of treatment and LNCaP cells after 24 h of treatment are reported in Appendix A for crystal violet staining and in Appendix A for the MTT assay. The effect of *Sm* (100 and 200 μg/mL) was also evaluated on 22Rv1 cells, with the data showing that the effect was statistically different in comparison with vehicle control (DMSO) and control (not treated) cells only after 72 h of treatment (Appendix A).

Fuco at increasing concentrations (0.5, 1, 5, 10, and 50 µg/mL) was tested on DU145 and LNCaP cells and was active in inhibiting PCa cell proliferation. After 48 and 72 h of treatment, it decreased cancer cell viability in a statistically significant manner compared to control cells and the vehicle control for both DU145 (Figure 2A) and LNCaP cells (Figure 2B). Subsequently, the same experiment on DU145 and LNCaP cells was performed adding a Fuco concentration of 100 µg/mL. Effects at 72 h at concentrations above 5 µg/mL were confirmed (Appendix A). Crystal violet assay was performed on 22Rv1 cells treated with Fuco at 20 and 100 µg/mL; both concentrations showed a significant decrease in cell proliferation at 48 and 72 h (*p* ≤ 0.0001, Appendix A)

### 3.2. Identification of Apoptosis on DU145 Cells

We tested whether the reduced proliferation of DU145 PCa cells following treatment with *Sm* extract and Fuco was associated with the induction of apoptosis. We found that *Sm* at 100 µg/mL was able to induce early apoptosis and 200 µg/mL could induce late apoptosis at 24, while both concentrations induced late apoptosis at 48 h (Figure 3). Treatment with Fuco, instead, did not induce apoptosis at 24 h at either concentration (Figure 3). Following 48 h of treatment, Fuco started to induce apoptosis only at 100 µg/mL (Figure 3). Similar apoptotic effects of *Sm* and Fuco were also observed on the 22Rv1 PCa cell line (Appendix A).

Based on these and previous results, we selected 100 µg/mL for *Sm* and 20 µg/mL for Fuco for further studies.

### 3.3. Effect of S. marinoi Extract and Fucoxanthin on Vascular Mimicry of PCa Cells

*Sm* (100 µg/mL) and Fuco (20 µg/mL) treatments on DU145 cells plated on Matrigel were able to inhibit interconnected structures, similar to capillaries, which has been defined as vascular mimicry (Figure 4A). Treatment with *Sm* at 100 μg/mL significantly reduced network formation compared to the FBS stimulus. The same results were obtained when DU145 cells were treated with Fuco at 20 µg/mL. DMSO induced no effects at the corresponding dilutions of *Sm* and Fuco concentrations (Figure 4B).

### 3.4. Fucoxanthin Potently Inhibits Cell Migration

The directional migration of endothelial cells represents the crucial event of angiogenesis. In the Boyden chamber and transwell migration assays, we examined cell migration in the absence or presence of Fuco (20 µg/mL). Fuco showed a significant inhibitory effect on HUVEC migration (Figure 5). Treatment with Fuco also significantly decreased HUVEC migration in the scratch assay (Appendix A).

In the wound healing assay, we also examined PCa cell migration in the mechanical scratch wound in the absence or presence of Fuco. Images of scratch areas from the timepoints 0, 24, and 48 h are illustrated in Appendix A, which also shows the representative control at each timepoint, indicating that the scratch was half closed within 24 h and completely closed after 48 h.

### 3.5. Gene Expression Profiling in Prostate Cancer Treated with S. marinoi Extract and Fucoxanthin

We investigated the potential effects of *Sm* and Fuco on the gene expression of DU145 cells by qPCR with custom-made primers after 6 and 24 h treatment. Cells treated with *Sm* (100 μg/mL) and Fuco (20 µg/mL) showed a downregulation of inflammation- and angiogenesis-associated genes (Figure 6). The most significantly downregulated genes after treatment with both *Sm* and Fuco pigment were CXCR4, MMP9, TIMP1, IL10, and IL6 at 6 h and TGFβ1 and IL10 at 24 h (Appendix A). Fuco decreased the expression of TGFβ1, TIMP2, TNFα, and STAT3 after 6 h in statistically significant manner; the expression of TGFβ2 both at 6 h and 24 h; and the expression of CXCR4 and IL6 after 24 h. *Sm* treatment did not modulate the gene expression of TGFβ1, TGFβ2, TNFα, TIMP2, and STAT3 after 6 h of treatment. CXCL8 and VEGF genes at 6 and 24 h of treatment and MMP9, TIMP1, TIMP2, TNFα, and STAT3 genes at 24 h of treatment were not statistically significantly differentially expressed with both *Sm* and Fuco treatments compared to the control (Appendix A).

### 3.6. Effect of S. marinoi Extract and Fucoxanthin on HUVEC Proliferation

Given the decreased expression of pro-angiogenic genes by *Sm* (Figure 7) and Fuco treatments (Figure 8) observed in PCa cells, we studied the effect of the extract and Fuco on endothelial cells. Overall, HUVECs showed slower proliferation than PCa cells given the lower concentrations and higher proliferation of HUVECs. At 72 h, the treatment with *Sm* (100 µg/mL) was significantly more effective in reducing cell proliferation with respect to the control, both in the crystal violet and MTT assays (*p* ≤ 0.0001 and *p* ≤ 0.05, respectively; Figure 7 A, B). Treatment with Fuco at concentrations higher than 5 µg/mL were more effective in reducing cell number than the control and vehicle control, both at 48 and 72 h, in the crystal violet assay (*p* ≤ 0.0001; Figure 8).

### 3.7. S. marinoi Extract and Fucoxanthin Effects on HUVEC Morphogenesis

We investigated the ability of *Sm* and Fuco to inhibit endothelial cell capillary-like structures on Matrigel. We assessed if *Sm* or Fuco could interfere with the ability of HUVECs to induce epithelial morphogenesis in a functional way (Figure 9). We observed that medium with FBS induced capillary-like network formation on a Matrigel layer, and treatment with *Sm* (50, 100, and 200 μg/mL) or Fuco (50, 100, and 200 µg/mL) significantly reduced this capability (Figure 9A). This effect is shown in Figure 9B by the quantification of the number (Nb) and total length of master segments and total mesh area. These results suggested that *Sm* and Fuco could be considered as potential angiogenesis inhibitors by reducing capillary-like morphogenesis of HUVECs in vitro.

### 3.8. Gene Expression Profiling in HUVECs Treated with S. marinoi Extract and Marine Pigment Fucoxanthin

We investigated the potential effects of *Sm* and the marine pigment Fuco on gene expression in an inflammation model of HUVECs induced by TNFα (10 ng/mL) and EtOH as control (Figure 10). *Sm* (100 μg/mL) and Fuco (20 µg/mL) treatment showed a statistically significant decrease in the expression of CXCL8, TGFβ1, TGFβ2, VEGF, MMP9, TIMP1, TIMP2, VCAN (Versican), and STAT3 genes. The IL6 gene was downregulated after treatment with Fuco (20 µg/mL) (*p* ≤ 0.0001) while CXCR4 showed a trend of being downregulated after 6 h of treatment with *Sm* (*p* ≤ 0.0001).

### 3.9. Antibody Arrays

We investigated the effect of Fuco on release of angiogenic factors by antibody array, using CM collected from stimulated HUVECs and DU145 cells. The visualized spots showed positive detection of the cytokines. Each cytokine detection was performed in duplicate.

In HUVECs, Fuco strongly downregulated Angiopoietin 2, Angiogenin, and CXCL5, while PLGF, TIMP1, TIMP2, MMP1, IL6, and PDGF-BB were sensibly downregulated (Figure 11 and Appendix A). In DU145 cells, Angiogenin, CXCL5, GRO, PDGF-BB, TIMP1, TIMP2 Angiopoietin 2, and IL1β appeared to be sensibly downregulated (Appendix A).

## 4. Discussion

Recent studies highlighted the anti-proliferative, anti-inflammatory, and anti-cancer effects of marine microalgae and bioactive compounds contained therein [34,65]. Among them, some pigments are exclusively aquatic, i.e., not retrieved in terrestrial plants. This is the case of the xanthophyll Fuco, which is present in chlorophyll c (chl c)-containing microalgae and brown macroalgae. Fuco is probably the most explored molecule from brown algae [75,76] due to its recognized effects on human health [77,78], including mainly anti-diabetic or anti-obesity effects but also anti-oxidant, anti-cancer, and anti-inflammatory effects. Furthermore, Fuco has been shown to have protective effects in hepatic, cardiovascular, renal, respiratory, metabolic, and skin diseases [79,80]. Fuco has started to be exploited and can be found in some products available on the market [81,82]. Fuco is also directly consumed through brown algae by humans [83].

However, it is important to point out that macroalgal exploitation is much less eco-sustainable than that of microalgae, except when considering beached macroalgal biomass [84]. One of the most open challenges of microalgal sources is retrieving bioactive molecules or extract/biomass with benefits for human health for consumption as dietary supplements or nutraceuticals [85]. The actual drawback is the production cost, which is still too high to make microalgal derivatives an affordable source of wellness for humans.

Among microalgae, diatoms are greatly attractive since they are generally fast-growing species with great cell size or biovolume and present high physiological plasticity, allowing to grow them under different conditions. Fuco is the main accessory pigment (carotenoid) in diatoms and its content can reach more than 40 mg × g^−1^ dry weight [76], with values generally higher than those reported in macroalgae.

All of the information supports the idea of an industrial-based, conceivable exploitation of microalgal Fuco. It is therefore important to deeply investigate its bioactivity. Here, we focused on its chemopreventive and anti-angiogenic activities that we compared with the total diatom extract. It is noteworthy that the Fuco content—as for all carotenoids—in photosynthetic organisms is greatly variable and depends on many factors, with light being the major forcing factor. *S. marinoi* displays a content of Fuco ranging between 40 and 290 fg/cell depending on the light climate [32,86].

Fuco inhibits proliferation in human cancer cells by decreasing the expression of angiogenic factors, including VEGF [72]. Fuco modulates GADD45 in PCa DU145 and LNCaP cells, which induces G1 cell cycle arrest [87,88]. Fuco causes apoptosis in PC-3, DU 145, and LNCaP cells [89,90].

Our data showed that an *Sm* extract and Fuco significantly inhibited the expression of pro-angiogenic factors. Levels of tissue inhibitors of metalloproteinases (TIMPs), although downregulating the activity of MMPs [91,92,93], are often elevated in cancer. For this reason, the angiogenic response can be the result of the modulation of invasion-associated genes through the balance between pro- and anti-angiogenic factors. *Sm* and Fuco hindered the vascular mimicry of prostate cancer cells on a basement membrane gel. We also demonstrated the inhibition in vitro of capillary-like structure formation when endothelial cells were treated with *Sm* and Fuco. These data confirmed the anti-angiogenic activity of *Sm* and its pigment, as also observed by Jang et al. 2021 [94]. In particular, we evaluated, at the molecular level, the downregulation of pro-inflammatory chemokines and cytokines that are directly responsible for angiogenesis promotion during tumor progression [95,96,97]. We have studied similar effects with other supplements, such as acetyl-L-carnitine and polyphenol-rich extract from olive mill wastewater [98,99].

Gene and protein expression were evaluated by qPCR and antibody array on secreted products from HUVECs and PCa cells. Altogether, we detected that *Sm* and Fuco significantly inhibited PCa cell viability and directly promoted their apoptosis, reduced the migration and capillary-like structure formation capacity of HUVECs, and decreased the expression of different angiogenic and inflammatory factors. Herein, we used PCa DU145 cells, which are more aggressive than LNCaP cells, and we observed that *Sm* and Fuco downregulated STAT3 gene expression in a statistically significant manner (*p* ≤ 0.0001). STAT3 is one of the critical members of the signal transducer and activator of transcription family. STAT3 has been shown to have a key role in fundamental cellular processes such as inflammation, cell growth, proliferation, differentiation, migration, metastasis, and apoptosis [100,101]. Our results found a parallelism with the effect of astaxanthin, another natural pigment produced by microalgae, in inhibiting STAT3 [102,103], but further and in-depth studies are needed to investigate the molecular mechanism involved in this inhibition effect together with other participating genes in the related pathways.

The present study investigated the potential chemopreventive and angiopreventive effects of *Sm* and Fuco, and we observed that these natural substances were able to inhibit proliferation, promote apoptosis, and could also suppress vascular mimicry formation in PCa cells. Moreover, they were found to reduce the migration ability and expression of genes involved in inflammatory and angiogenic processes in TNFα-stimulated HUVECs.

## 5. Conclusions

The marine environment represents a significant reservoir for the discovery of new molecules of pharmacological interest because of its very high chemical and biological diversity in secondary metabolites.

Our in vitro study encourages further investigation of marine microalgae and their bioactive compounds, suggesting that diatoms such as *Skeletonema marinoi* could be a promising pharmaceutical supplement as a “phytocomplex” thanks to their high content of fucoxanthin. Pending further investigations, *Sm* and fucoxanthin could be natural product candidates as potentially preventive for PCa owing to their anti-cancer and anti-angiogenic activities.

## Figures and Tables

**Figure 1 cells-12-01053-f001:**
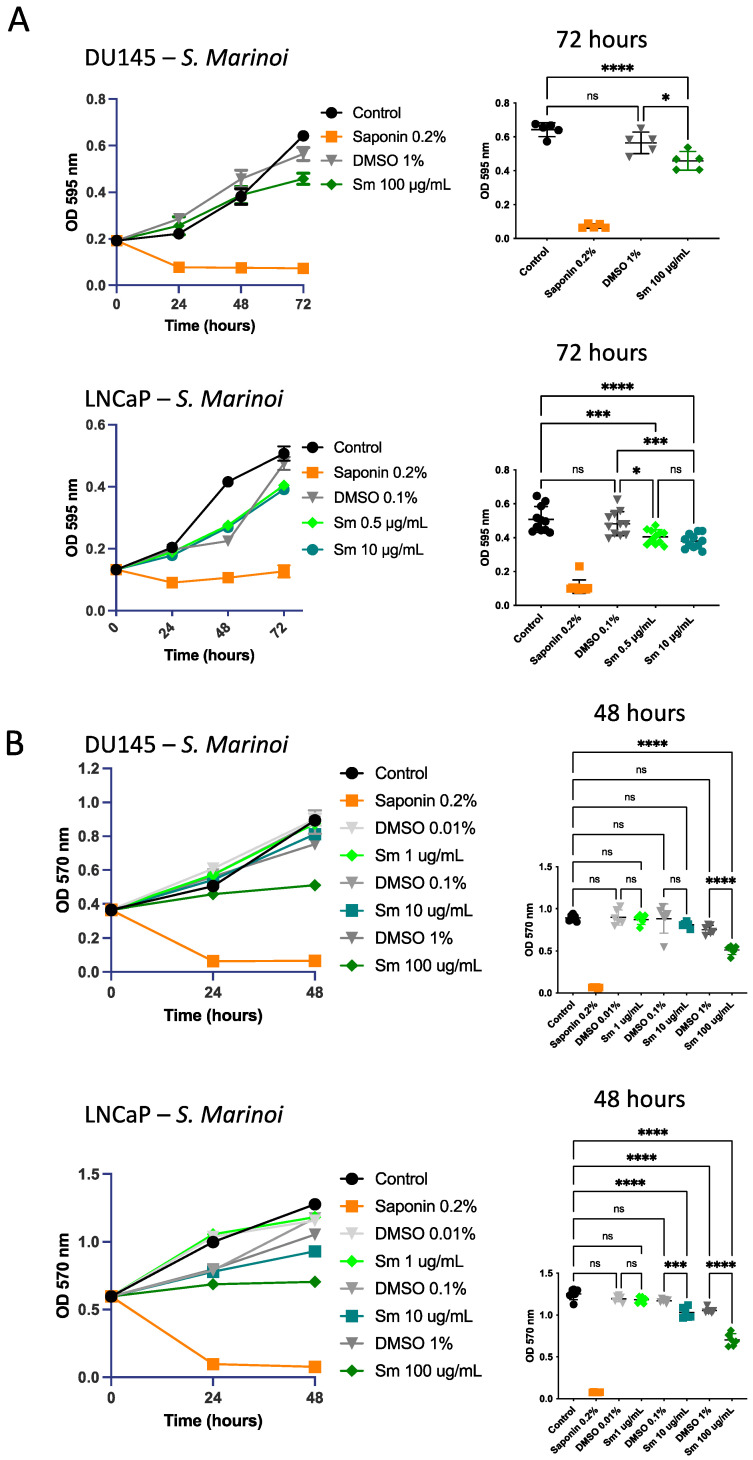
Effects of *S. marinoi* extract on prostate cancer (PCa) cell line proliferation. The proliferation rate was measured by (**A**) crystal violet assay (OD 595 nm) at 72 h and (**B**) MTT (OD 570) at 48 h. DU145 cells were treated with *S. marinoi (Sm)* at 1–10–100 μg/mL, LNCaP cells were treated with *Sm* at 0.5, 1, 10, and 100 µg/mL. Results are shown as mean ± SEM, one-way ANOVA, ns = not significant, * *p* < 0.05, *** *p* < 0.001, **** *p* < 0.0001 (*n* = 6).

**Figure 2 cells-12-01053-f002:**
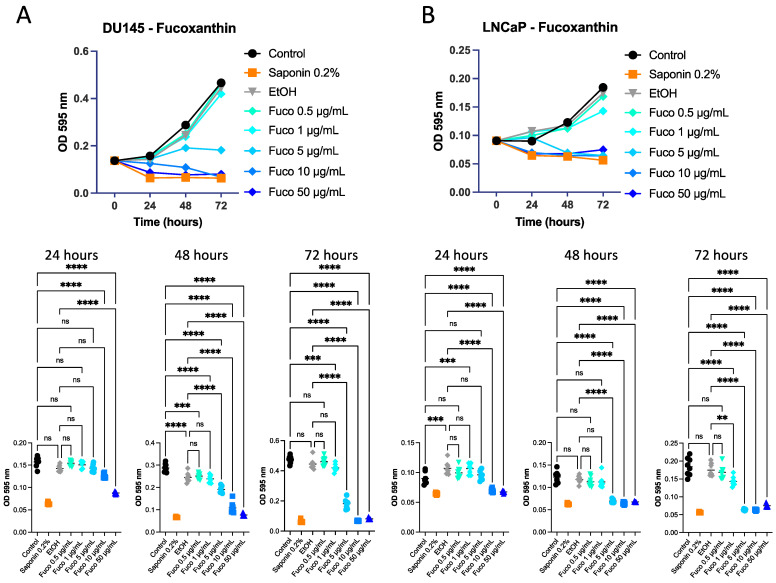
Effects of fucoxanthin pigment on PCa cell line proliferation. DU145 (**A**) and LNCaP (**B**) cells were treated with fucoxanthin (Fuco) at 0.5, 1, 5, 10, and 50 µg/mL for 24, 48 and 72 h. The proliferation rate was measured by crystal violet assay (OD 595 nm). Results are shown as mean ± SEM, one-way ANOVA, ns = not significant, ** *p* < 0.01, *** *p* < 0.001, **** *p* < 0.0001 (*n* = 8).

**Figure 3 cells-12-01053-f003:**
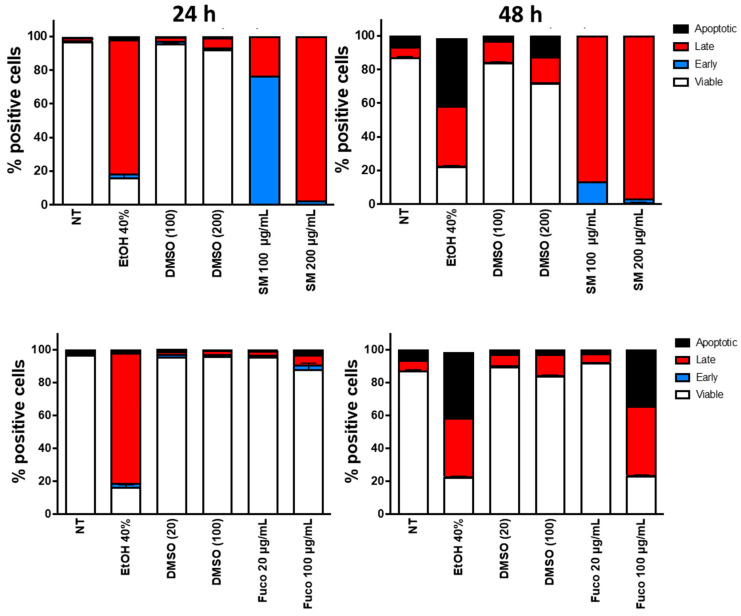
Effects of *S. marinoi* extract and fucoxanthin in vitro on the induction of apoptosis in DU145 PCa cell line. Cells were treated for 24 and 48 h with *Sm* (100 and 200 µg/mL) and Fuco (20 and 100 µg/mL). The effects on induction of apoptosis were evaluated by flow cytometry (FACS analysis) following labeling with Annexin V/PI. Data are represented as bar graphs (*n* = 2). Control (not treated cells), EtOH (ethanol used at 40% as positive control), and DMSO (vehicle control used to dissolve *Sm* and Fuco).

**Figure 4 cells-12-01053-f004:**
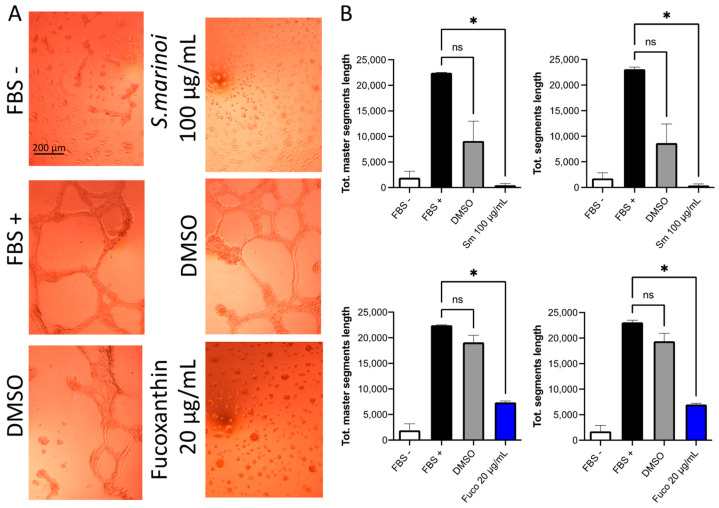
Effects of *S. marinoi* extract and fucoxanthin on DU145 tumor cell vascular mimicry. DMSO, at the corresponding dilutions, was used as vehicle control for both *Sm* (100 µg/mL) and Fuco (20 µg/mL). (**A**) Photographs of vascular mimicry structures were taken at 10× magnification, representative images are shown, scale bar: 200 µm. (**B**) Bar graphs are shown (ImageJ software and Angiogenesis Analyzer tool). Data are shown as mean ± SEM, one-way ANOVA, ns = not significant, * *p* < 0.05; vs. FBS+ (*n* = 4).

**Figure 5 cells-12-01053-f005:**
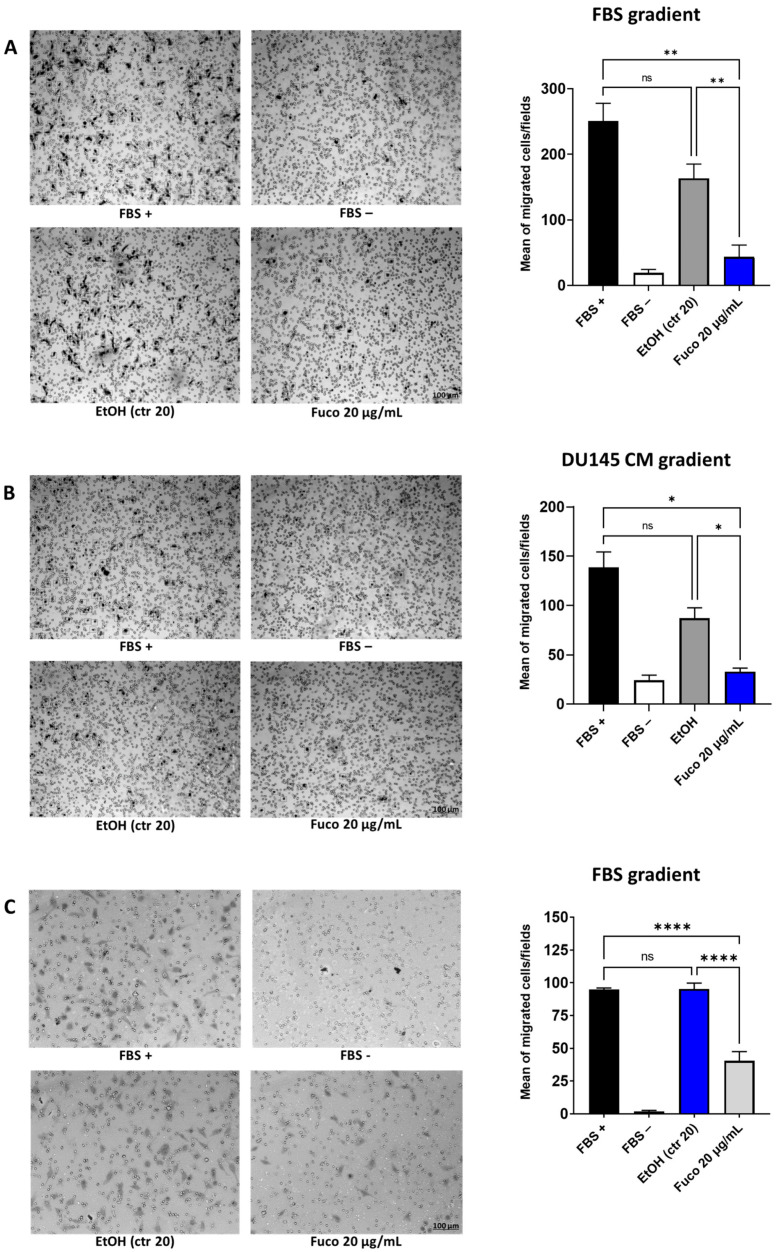
Effects of fucoxanthin on HUVEC migration. Cells were treated with Fuco (20 µg/mL) or EtOH as the vehicle control, for 6 h. Cell migration was examined using the Boyden chamber assay with FBS (**A**) and DU145 CM (**B**) as chemoattractants and using the transwell assay (**C**) with FBS as a chemoattractant. Photographs of filters were taken at 10× magnification, representative images are shown, scale bar: 100 µm. Migration was quantified by counting the number of migrated cells in three independent fields for each experimental condition and data are represented as the mean of the four means calculated for each condition (ImageJ software). Bar graphs are shown, results are shown as mean ± SEM, one-way ANOVA, ns = not significant, * *p* < 0.05, ** *p* < 0.01, **** *p* < 0.0001 vs. FBS+ (*n* = 4).

**Figure 6 cells-12-01053-f006:**
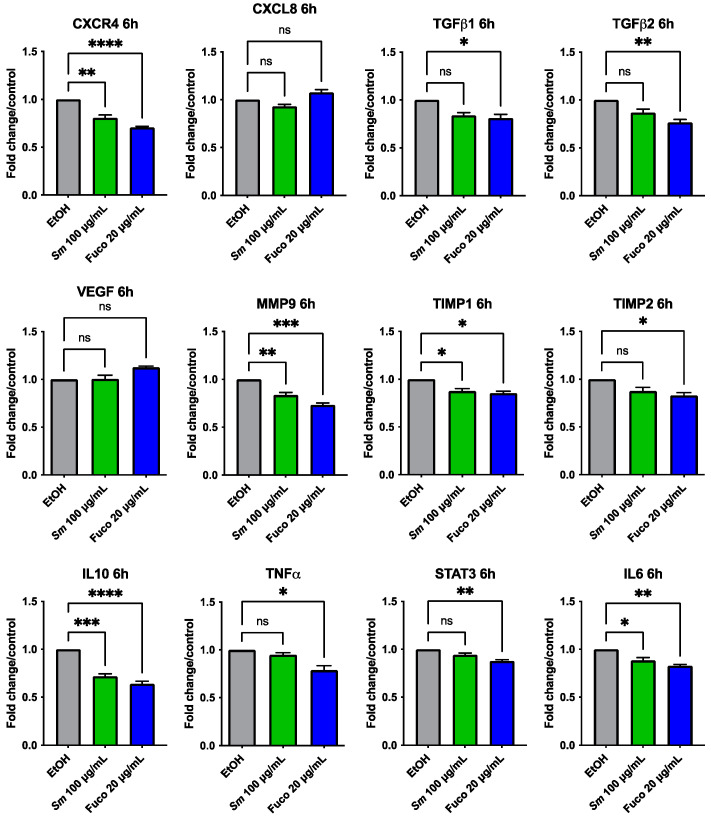
Gene expression profiling in PCa cell treated with *S. marinoi* and fucoxanthin. The ability of *Sm* (100 µg/mL) and Fuco (20 µg/mL) to inhibit gene expression in DU145 cells was determined by qPCR following 6 h of treatment. Data are shown as relative mRNA expression normalized to β-actin, results are shown as mean ± SEM, one-way ANOVA, ns = not significant, * = *p* < 0.05, ** = *p* < 0.01, *** = *p* < 0.001, **** = *p* < 0.0001. (*n* = 6).

**Figure 7 cells-12-01053-f007:**
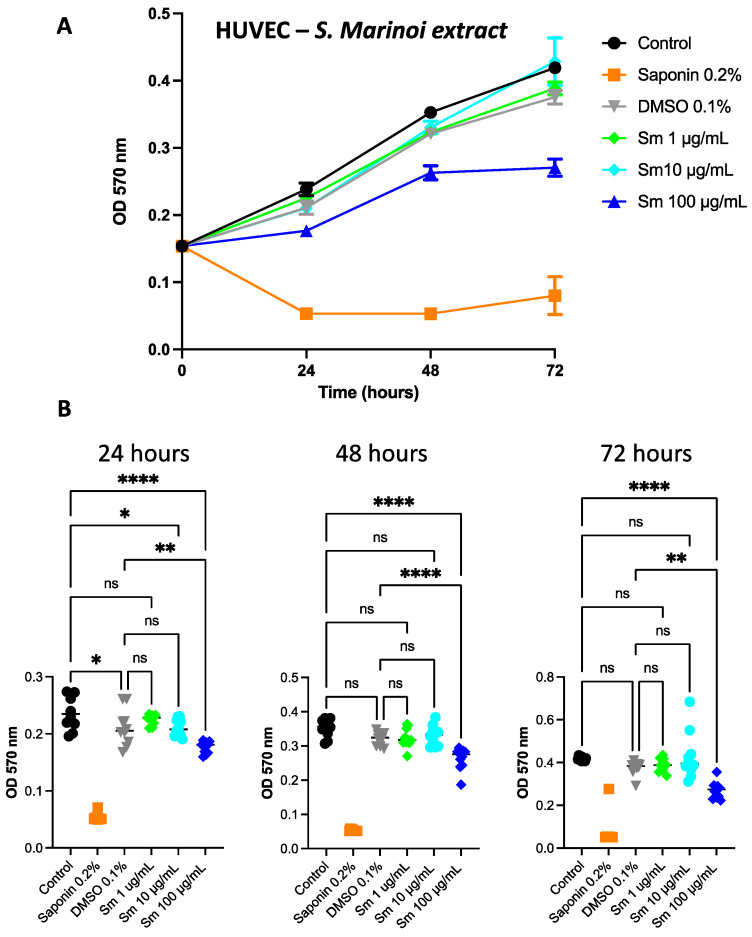
Effects of *S. marinoi* extract on HUVEC proliferation. (**A**) The proliferation rate was measured by MTT assay (OD 570 nm). (**B**) HUVECs treated with *Sm* at 1, 10, and 100 µg/mL for 24, 48, and 72 h and corresponding control and DMSO concentrations. Results are shown as mean ± SEM, one-way ANOVA, ns = not significant, * *p* < 0.05, ** *p* < 0.01, **** *p* < 0.0001 (*n* = 10).

**Figure 8 cells-12-01053-f008:**
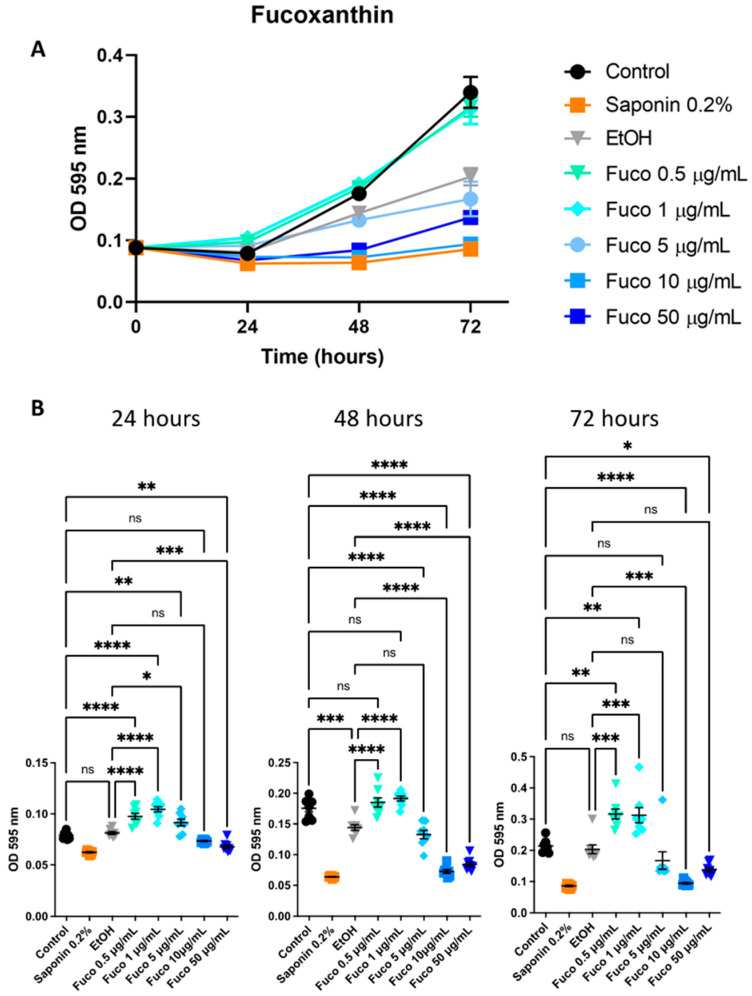
Effects of fucoxanthin on HUVEC proliferation. (**A**) The proliferation rate was measured by crystal violet assay (OD 595 nm) on HUVECs treated with Fuco at 0.5, 1, 5, 10, and 50 µg/mL and corresponding EtOH concentrations, for 24, 48, 72 h. (**B**) Results are represented as mean ± SEM, one-way ANOVA, ns = not significant, * *p* < 0.05, ** *p* < 0.01, *** *p* < 0.001, **** *p* < 0.0001, (*n* = 8).

**Figure 9 cells-12-01053-f009:**
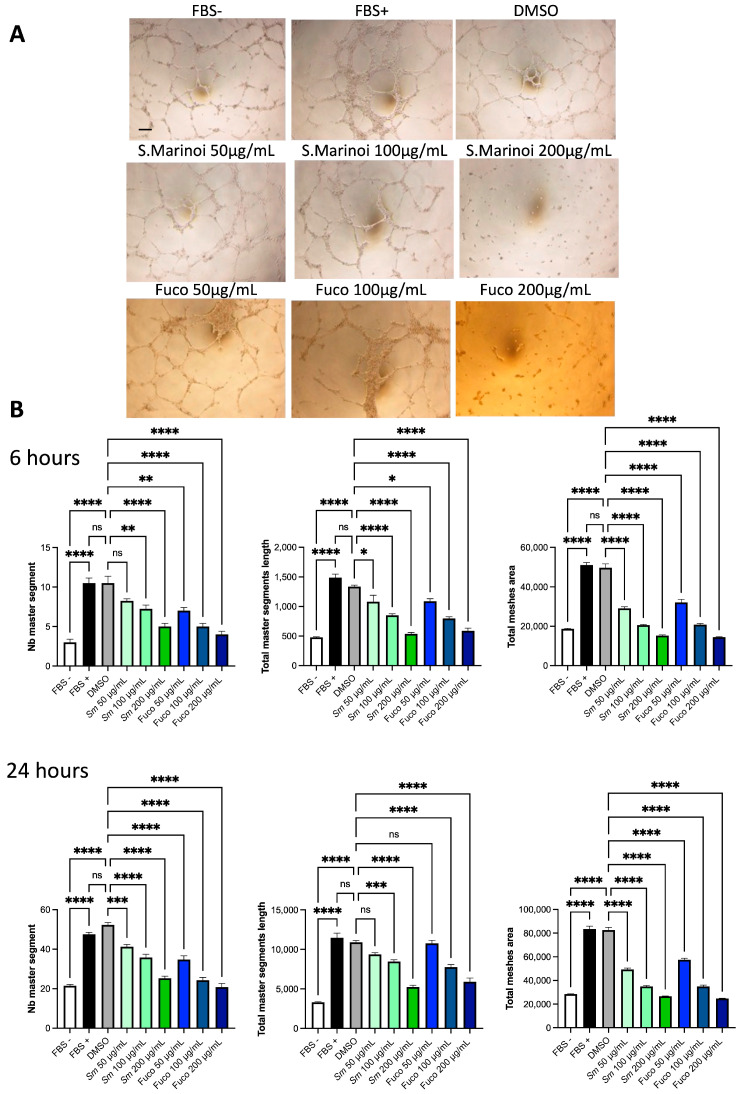
Effect of *S. marinoi* extract and fucoxanthin on capillary-like structure formation of HUVECs. Cells were placed in 96-well plates coated with a layer of Matrigel (2.5 × 10^3^ per well) and treated with *Sm* and Fuco (50, 100, or 200 µg/mL) or incubated with vehicle alone for 6 and 24 h. FBS -/-: cells cultured in serum-free EBM-2 medium, FBS +/+: cells grown in complete EBM-2 medium, were used as the negative and positive controls, respectively. (**A**) Microphotographs were taken at 5× magnification, representative images are shown, scale bar 200 µm. (**B**) Images were quantified by Angiogenesis Analyzer ImageJ tool kit. The number (Nb) of master segments as well as the total master segment length and total meshes area were evaluated. The effect of *Sm* and pigment on HUVECs’ ability to form capillary-like structures on Matrigel was compared to vehicle-treated cells. Data are shown as mean ± SEM, one-way ANOVA, ns = not significant, * *p* < 0.05, ** *p* < 0.01, *** *p* < 0.001, **** *p* < 0.0001. (*n* = 4).

**Figure 10 cells-12-01053-f010:**
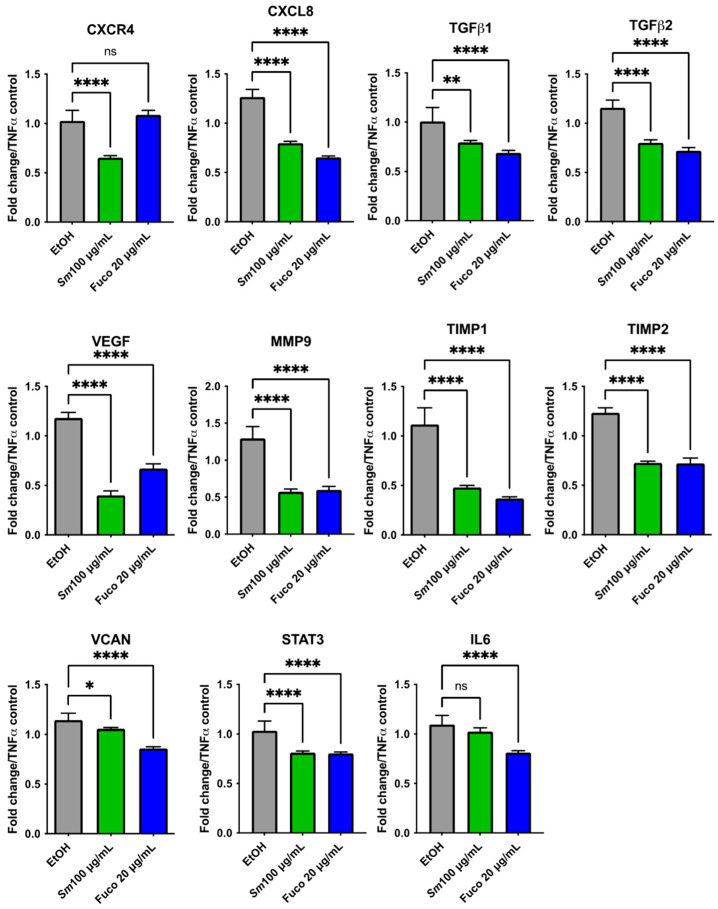
Gene expression profiling in endothelial cells treated with *S. marinoi* and marine pigment fucoxanthin. The ability of *Sm* (100 µg/mL) (**A**) and Fuco (20 µg/mL) (**B**) to inhibit cytokine expression in HUVECs stimulated with TNFα (10 ng/mL) was determined by qPCR following 6 h of treatment. Data are shown as relative mRNA expression normalized to β-actin and TNFα-stimulated control, EtOH (vehicle control), mean ± SEM, one-way ANOVA, * *p* < 0.05, ** *p* < 0.01, **** *p* < 0.0001, (*n* = 9).

**Figure 11 cells-12-01053-f011:**
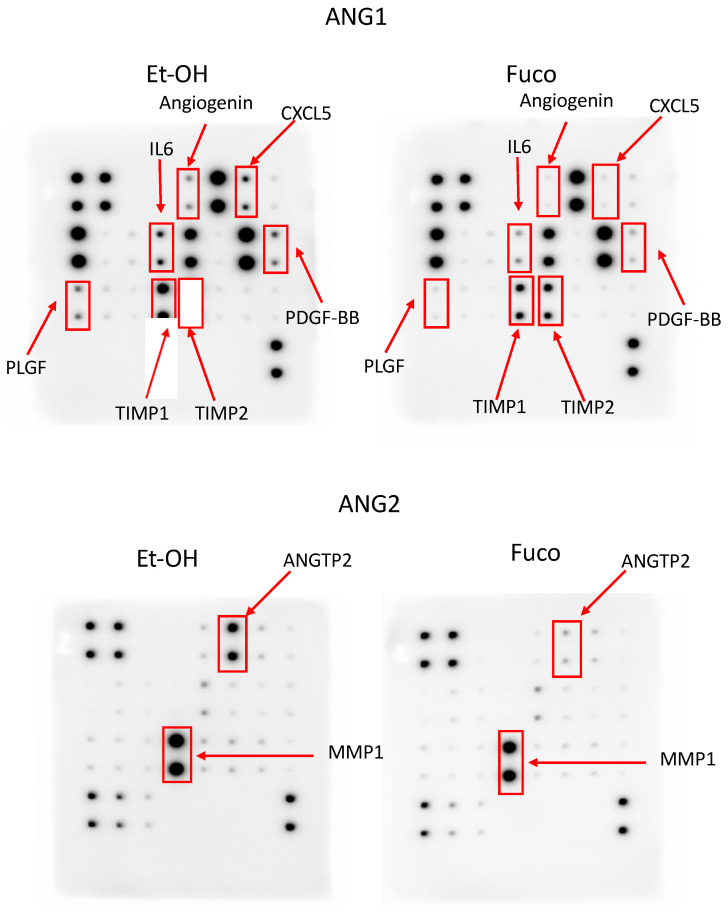
Protein expression profiling in endothelial cells treated with fucoxanthin. Antibody arrays representing cytokine expression in HUVECs stimulated with TNFα (10 ng/mL) after treatment with Fuco (20 µg/mL). EtOH was used as vehicle control (*n* = 2). ANG1 and ANG2: Human Angiogenesis Array C1 and C2, respectively.

## Data Availability

The authors confirm that the data supporting the findings of this study are available within the article and its Appendix A. Further data that support the findings of this study are available on request from the corresponding authors [A.A. and L.C.].

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
