# Peer review of "Skeletonema marinoi Extracts and Associated Carotenoid Fucoxanthin Downregulate Pro-Angiogenic Mediators on Prostate Cancer and Endothelial Cells"

_cells, 2023, doi:10.3390/cells12071053_

Round 1

Reviewer 1 Report

This is an interesting paper which provides novel dietary approach in prostate cancer prevention. The authors propose that SM extract and Fucoxanthin were able to inhibit PCa cell growth and inhibited vascular network formation on endothelial cells and vascular mimicry. Overall, the manuscript is well thought and the approaches are valid. However, the author's conclusions should be supported by identification of molecular mechanisms. The results should be better written as they are a bit confusing especially in the initial part. Please carefully proofread the manuscript and correct several typos or mistakes.

Author Response

We thank the reviewer for the positive comment “the manuscript is well thought and the approaches are valid”.

To answer his comment “results should be better written” we are extensively working on expanding and better explaining the results.

Reviewer 2 Report

For authors,

The marine-derived biomass would be an elusive and attractive target to intend to show the anti-cancer and anti-angiogenic effects on prostate cancer. Moreover, the authors present the combined effect of conventional chemos with in vitro based on experimental results. However, unfortunately, the methodology is outdated and tenuous for studying food and functional food function. For example, it is impossible to claim or conclude that the extracts exhibit anti-cancer and anti-angiogenic effects with only MTT and tube formation assays, respectively. Furthermore, insufficient experimental evidence, such as migration, wound/scratch assays, aortic ring assay, underlying molecular mechanism, and in vivo analysis, are highly concerning. Therefore, the authors need to add substantial amounts of the experimental result to convince readers of the authors' claim. 

There are many concerns about marine-derived biomass properties and experimental and editorial issues that authors need to address.

First, it should mention the editorial issues. The manuscript is seemingly organized but is not polished. There are numerous typos, and the authors utilize inconsistent abbreviations such as Sm/SM, HUVEC/HUVE, and Fuco, potentially misleading readers. It might be admirable to show statistical differences among all groups in the figures with the large symbol. Still, the main contents of the figures, line and column graphs, are small to have difficulty in seeing the results, which would again be the potential risk to misguided the reader. All microscopic pictures have no scale bar. The figure legend is missing information for the replicate numbers and the mean. A figure is missing the symbol for statistics and uses the wrong style for super-and subscript. It would be better to consider the color choice of figures for the color blind readers. All editorial errors should be rectified. 

The one main issues are to show a lack of information, not literature, on this main Skeletonema marinori (SM)The marine products must be mixers of a variety of nutrients. Authors have to show the result of the components in the SM. Furthermore, there is no content information on the biomass extract. Mechanical and chemical extraction would not give us the same results every time. How to standardize the batch effect? It is unclear why and how the author concludes that fucoxanthin in the SM extract is the pivotal active component in exhibiting the effects. How do authors decide the dosage? Authors need to explain the rationale based on experimental results. As the authors described, numerous studies of fucoxanthin on the anti-cancer effects are available, and this study looks redundant to other published papers if authors focus on fucoxanthin as the only active component in the SM. What are the apparent differences between this study and other studies? 

For experiments, the introduction's explanation is not convincing why the authors target prostate cancers for SM. Why not testicle cancer, cervical cancer, or other types of deadliest cancer? 

The authors tested the SM effect on only two prostate cancer cell lines, though using HUVEC, it is not enough to generalize the SM effect on the prostate cancer cells. Authors need to examine at least more cell lines. 

It is again unclear why authors use saponin as a positive control. Saponins are known to cause apoptosis in cancer cells. What is the scientific rationale for choosing the compound as this assy control? Do authors intend to say that SM does not induce apoptosis in prostate cancer cells? The authors have to show experimental evidence through another assay. 

Why and how does SM suppress prostate cancer's proliferation? Why does SM induce the effect in 72 hours, not 48 hours? What has happened to the treated cells in the 24 hours? 

What is the leading cause of the anti-proliferation effect of SM on prostate cancer cells? It seems to show different mechanisms to suppress the proliferation of prostate cancer cells between SM and fucoxanthin. How do the authors explain the reason for the differences in the effect of prostate cancer? And what is the molecular mechanism of the effect? 

Tube formation assay is only part of the in vitro analyses for angiogenesis. Authors should perform more assessments, such as wound/scratch, migration assay, and ex vivo aortic ring assay, to claim the effect. 

It is still unconvincing that authors conclude the enhancement effect with the chemos. What is the mechanism behind it? 

What is the experimental rationale for treatment time for cytokine production? Why 6 hours? And what happens to cytokine production if the time is prolonged? What is the underlying molecular pathway to down-regulate cytokine production? 

Author Response

We like to thank the reviewer for the deep analysis and thorough revision work. We found the comments very useful and we profoundly edited the manuscript, with new experiments, new figures and more supplemental materials and edited text.

Major changes that we introduced are summarized below in 17 points:

  1. We explained better the rationale to study prostate cancer cells for chemoprevention (lines 107-126).
  2. The quantity of major pigments diatoxanthin and fucoxanthin is now shown in supplementary materials Table 1. Fucoxanthin was the highest in concentration and was chosen as most representative. This is now stated in the manuscript.
  3. The amount of pigments was similar for three different batches of algal extract (shown in the supplementary material). For experiments, we run a range of fucoxanthin concentrations compatible with its representation in the diatom, and then for pathways studies we used the concentration that appeared most relevant to the biological effects.
  4. The extraction procedure has been set up after a comparative analysis of different methodologies and used since many years (e.g., Smerilli et al., 2017, 2019). In this case, the 2 min lasting mechanical extraction was the best to harvest carotenoids. This is now stated in the manuscript.
  5. Since methanol allows full extraction of carotenoids and that fucoxanthin was the highest concentrated compound in this family, it is supposed that fucoxanthin is a highly bioactive compound (ref Pistelli et al. (2021)), and that it is a compound responsible for at least part of the bioactivity property of the marinoi extract. For this and for future supplement studies, we decided to use pure fucoxanthin too.
  6. We performed experiments on a minimum of two prostate cancer cell lines (DU145 and LNCaP), which is the standard requested by most journals (although we find several publications also on MDPI on just one cell line) for instance CIT astaxanthin prostate. In any case, to increase cell lines we added the results obtained on 22Rv1 prostate cancer cell line in supplementary figures. We performed both proliferation (Figure S3, S5) and apoptosis assays (Figure S7) were performed on this 3rd cell line.
  7. We tested different concentrations of Sm and pigment on all cell lines for dosages and then we selected the most significant representative for each one for experiments on mechanisms.
  8. We added the sentence to the paper: rational has been expanded, we added sentences (Lines 109-113): We agree that numerous studies of fucoxanthin on the anti-cancer effects are available. However, our study is now very detailed with new information. Furthermore, several studies describe the effects of purified molecules, while our study compares an extract of the diatom, marinoi as a very promising nutraceutical supplement as “phytocomplex”, also thanks to its content in fucoxanthin. We better specified this in our conclusions.

It is not surprising that somewhat different mechanisms to suppress the proliferation of prostate cancer cells appear between SM and fucoxanthin, since SM contains several molecules. In any case, pure fucoxanthin has a very effective action and it can be easily used as diet supplement. Therefore, our study suggests that diatoms, such as S. marinoi could be a very promising pharmaceutical supplement as phytocomplex, as well as pure fucoxanthin.

  1. We show more evident data for vitality at different times, such as 24, 48 and 72 hrs, which were present but maybe less visible. The effect exerted by Sm becomes significant after 48 h and it is evident after 72 h of treatment. We added figures S1 (crystal violet assay (OD 595 nm) 24, 48 hours) and S2 (MTT (OD 570 nm) 24 hours).

Saponin has been used as a control for cell viability as in Gallazzi et al 20201, and we added the results obtained in apoptosis assay, as questioned by the reviewer.

  1. We added some of the techniques suggested by the reviewer, such as:
  2. A) migration in Boyden chambers (chemotaxis)
  3. B) wound/scratch migration assay,

and added those results in new figures (A Fig. 5, B Fig. S7 and Fig. S8).

  1. Cytokine expression can be modulated rapidly, in 1 hour, many of them peak between 3-6 hours. Therefore, we chose 6 h as a better time point to collect cells for RNA extraction.

We carried out also treatments of cells both for 6 and 24 hours and we confirmed that results are more significant after 6 h than after 24 h (Figure S9).

  1. Molecular effects were analysed through the qPCR and protein (antibody array) modulation data, and it is more deeply discussed.
  2. We show now antibody arrays data for both prostate and HUVEC cells.
  3. The reviewer states that “It is still unconvincing that authors conclude the enhancement effect with the chemos.” The paper is now very large with a thigh number of novel data; therefore, we convene not to consider discussing the additive effect with chemotherapy, which will be eventually analysed also for mechanism in a future study.
  4. Typos, and inconsistent abbreviations have been fixed.
  5. Scale bars have been added to microscopic pictures.
  6. The symbol for statistics has been added to the figure where it was missing.

1 Gallazzi M, Festa M, Corradino P, Sansone C, Albini A, Noonan DM. An extract of olive mill wastewater downregulates growth, adhesion and invasion pathways in lung cancer cells: involvement of CXCR4. Nutrients. (2020) 12:903. doi: 10.3390/nu12040903

Round 2

Reviewer 1 Report

The manuscript was extensively revised thus I retain that it could be accepted in the present form

Author Response

Thank you very much.

Reviewer 2 Report

Dear authors,

Evaluating their findings using toxicological and oncological in vivo models would still be highly recommended to claim that marine-derived biomass might be a pharmaceutical supplement. Therefore, it would be good to tune down their claim in lines 755-756 on page 36. However, the authors have performed extensive experiments in response to my questions, which is highly appreciated.

Author Response

Dear Reviewer

We have grateful for your time and effort in reviewing our manuscript.
The feedback has been invaluable in improving the content and presentation of the paper.

We modified the lines indicated by you, in attachment Track changes file with modifications, lines 611-617, page 23.